# A Local Temporal Difference Code for Distributional Reinforcement Learning

**Pablo Tano**[*]
Basic Neurosciences
University of Geneva

**Peter Dayan**
MPI for Biological Cybernetics
University of Tübingen

**Alexandre Pouget**
Basic Neurosciences
University of Geneva

## Abstract

Recent theoretical and experimental results suggest that the dopamine system implements distributional temporal difference backups, allowing learning of the entire distributions of the long-run values of states rather than just their expected values. However, the distributional codes explored so far rely on a complex imputation step which crucially relies on spatial non-locality: in order to compute reward prediction errors, units must know not only their own state but also the states of the other units. It is far from clear how these steps could be implemented in realistic neural circuits. Here, we introduce the Laplace code: a local temporal difference code for distributional reinforcement learning that is representationally powerful and computationally straightforward. The code decomposes value distributions and prediction errors across three separated dimensions: reward magnitude (related to distributional quantiles), temporal discounting (related to the Laplace transform of future rewards) and time horizon (related to eligibility traces). Besides lending itself to a local learning rule, the decomposition recovers the temporal evolution of the immediate reward distribution, indicating all possible rewards at all future times. This increases representational capacity and allows for temporally-flexible computations that immediately adjust to changing horizons or discount factors.

## 1 Introduction

In the traditional Reinforcement Learning (RL) framework, agents make decisions by learning and maximizing the *scalar values* of states, which quantify the expected sums of discounted future rewards that will be encountered from those states [1]. Recently, several results have suggested that humans and animals keep track of richer information about the distribution of future rewards, besides just its expectation [2–5]. Indeed, recent machine learning advances suggest that, besides providing more flexibility for decision-making, requiring the entire *value distribution* to be learned leads to representations that support improved average performance, since the agent needs to represent separately states with the same expected value but different value statistics [6–8].

These results naturally pose the question of how such distributional estimators are learned and represented by neural systems. A distributional RL algorithm called *Expectile temporal difference (TD) learning* [9] has been recently proposed as a neurally plausible method that extends the conventional temporal difference reward prediction error (RPE) theory of dopamine activity [10]. Expectile TD algorithms learn a set of estimators that converge to the expectiles of the value distribution. Importantly, as we explain in the next section, the Expectile algorithms are *non-local*, which is a critical problem when considering neurally-plausible implementations of distributional RL.

Here, we show that the non-locality of distributional TD algorithms is not required to learn the value distribution. We show that an ensemble of independent units performing traditional and *local* TD

---

[*]Corresponding author: `Pablo.TanoRetamales@unige.ch`

backups can recover the value distribution. Units in the ensemble have selectivities that vary along three dimensions: reward magnitude, temporal discount factors, and an explicit memory about past outcomes. In addition to the value distribution, it is possible to recover the *temporal evolution of the immediate reward distribution* from our code, by taking an inverse Laplace operator. The temporal evolution indicates the probability of obtaining different reward magnitudes at every timestep in the future. This additional information increases representational capacity and allows for a computation of expected and distributional value that immediately adapts to changes in the temporal horizon of the task. Finally, we illustrate a strong connection between our code and predictive representations, by showing that our code can be linearly computed from an ensemble of successor representations (SR) with different temporal discounts, a model recently proposed for the hippocampus [11].

## 2  Background

**Traditional RL: TD Learning**  Consider state evaluation in a Markov Process (MP) $(\mathcal{S}, \mathcal{P}, \mathcal{R})$, where $\mathcal{S}$ is the set of states, $\mathcal{R} : \mathcal{S} \to \mathbb{R}$ the reward function and $\mathcal{P}(s_{t+1}|s_t)$ the transition function. After observing the transition $s_t \to s_{t+1}$ and receiving a reward $r_t$, the TD algorithm performs the following backup on the value estimate of $s_t$:

$$V(s_t) \leftarrow V(s_t) + \alpha\delta(t) \tag{1}$$
$$\delta(t) = r_t + \gamma V(s_{t+1}) - V(s_t)$$

where $\alpha$ is the learning rate, $\gamma$ the temporal discount and $\delta(t)$ the RPE. Under fairly general conditions [12], this backup leads the value estimate to converge to the expected value of state $s$:

$$V(s) \to E\Big[\sum\nolimits_{\tau=0}^{\infty} \gamma^\tau r_{t+\tau}\Big|s_t = s\Big] \tag{2}$$

According to the traditional RPE dopamine theory [13, 14], the activity of dopaminergic neurons in the Ventral Tegmental Area (VTA) encodes the RPE, $\delta(t)$. Note that Eq. 1 is *local*, in the sense that, when there are multiple pairs of dopamine and value units (or population of units), each pair do not need to communicate with other pairs (Fig. 1a).

**Distributional RL: Expectile TD Learning**  In distributional RL, agents learn to approximate the distribution of values starting from state $s$: $P(\sum_{\tau=0}^{\infty} \gamma^\tau r_{t+\tau}|s_t = s)$. The Expectile TD algorithm consists of $N$ estimates. The TD backup for estimate $i$ is:

$$V_i(s_t) \leftarrow V_i(s_t) + \alpha_i^+ \delta_i(t) \ \text{ if } \delta_i(t) > 0 \tag{3}$$
$$V_i(s_t) \leftarrow V_i(s_t) + \alpha_i^- \delta_i(t) \ \text{ if } \delta_i(t) < 0$$
$$\delta_i(t) = r_t + \gamma\tilde{V}(s_{t+1}) - V_i(s_t)$$

where $\tilde{V}(s_{t+1})$ is a random sample from the value distribution of $s_{t+1}$. Estimate $V_i(s)$ converges to the $\tau_i = (\alpha_i^+)/(\alpha_i^+ + \alpha_i^-)$ expectile of the value distribution. For readers unfamiliar with *expectiles*, note that the 0.5 expectile is the mean (as the 0.5 quantile is the median). Recently, Dabney et al. [10] proposed that dopaminergic activity in the VTA corresponded to the $\delta_i$'s in Eq. 3.

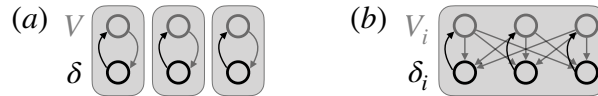

Figure 1: **(a)** Local TD circuit. **(b)** Non-local TD circuit required for learning an Expectile code. The units encoding the expectiles ($V_i$) must communicate stochastically with all the RPE units ($\delta_i$).

Importantly, in order for the backups to converge, the RPE $\delta_i(t)$ needs to receive the random sample $\tilde{V}(s_{t+1})$. To take this sample, the agent first needs to find a probability distribution consistent with the current set of expectiles $\{V_{1,\dots,N}(s_{t+1})\}$. This is called the *imputation step*, and consists of a high-dimensional, convex optimization problem that must be solved every time that dopamine neurons compute RPEs [9]. This code is therefore *non-local*, in the sense that each dopamine unit effectively needs to communicate with all other units, and must do so in a stochastic way through a complex imputation step (Fig. 1b). The non-local stochastic connectivity, in addition to the imputation step, makes the minimal circuitry required by the Expectile code significantly more complicated than under the traditional RPE dopamine theory. Although it is possible to modify the Expectile code to make it local (i.e. $\delta_i(t) = r_t + \gamma V_i(s_{t+1}) - V_i(s_t)$ in Eq. 3), this local version does not always converge to the correct value distribution, as shown in [9] and in the Appendix E.1 of this paper.

# 3 The Laplace code

## 3.1 TD learning with varying temporal discount

Consider state evaluation in the MP $(\mathcal{S}, \mathcal{P}, \mathcal{R})$ introduced in the previous section and an ensemble of traditional TD units with different temporal discounts $\gamma$. The backup on estimate $V_\gamma$ is:

$$V_\gamma(s_t) \leftarrow V_\gamma(s_t) + \alpha \delta_\gamma(t) \tag{4}$$

$$\delta_\gamma(t) = r_t + \gamma V_\gamma(s_{t+1}) - V_\gamma(s_t)$$

Under the same conditions as in traditional TD learning [12], estimate $V_\gamma(s_t)$ converges to:

$$V_\gamma(s_t) \rightarrow E\Big[\sum\nolimits_{\tau=0}^{\infty} \gamma^\tau r_{t+\tau}\Big|s_t\Big] = \sum\nolimits_{\tau=0}^{\infty} \gamma^\tau E[r_{t+\tau}|s_t] \tag{5}$$

This reveals a critical property: $V_\gamma(s_t)$ is the unilateral $Z$-transform of $E[r_{t+\tau}|s_t]$, with real-valued parameter $\gamma^{-1}$ (this is, the discrete-time equivalent of the Laplace transform). Since the $Z$-transform is invertible, in the limit of infinite $\gamma$'s we can recover $\{E[r_{t+\tau}|s_t]\}_{\tau=0}^{\infty}$ from $\{V_\gamma(s_t)\}_{\gamma \in (0,1)}$:

$$Z^{-1}\{V_\gamma(s_t)\}_{\gamma \in (0,1)} = \{E[r_{t+\tau}|s_t]\}_{\tau=0}^{\infty} \tag{6}$$

This insight generalizes to the continuous time limit. Using that $\gamma^\tau \equiv e^{-\tau(-\log \gamma)}$, Eq. 5 becomes:

$$V_\gamma(s_t) \rightarrow \int_0^\infty e^{-\tau(-\log \gamma)} E[r_{t+\tau}|s_t]\, d\tau \tag{7}$$

which shows that $V_\gamma(s_t)$ is the Laplace transform of $E[r_{t+\tau}|s_t]$ with parameter $-\log \gamma$. Therefore:

$$\mathcal{L}^{-1}\{V_\gamma(s_t)\}_{\gamma \in (0,1)} = \{E[r_{t+\tau}|s_t]\}_{\tau > 0} \tag{8}$$

Thus the converging points of the TD backups in Eq. 4 (i.e. $\{V_\gamma(s_t)\}_\gamma$) encode not only the expected sum of discounted rewards, as in traditional RL, but also the *expected reward at all future timesteps* (i.e. $\{E[r_{t+\tau}|s_t]\}_\tau$), though the latter lies in a different space, analogous to the frequency and temporal spaces of the Fourier transform. We name the two spaces $\gamma$-*space* and $\tau$-*space*, such that applying $\mathcal{L}^{-1}$ (or $Z^{-1}$) to the $\gamma$-space returns the $\tau$-space.

Importantly, for a finite ensemble of N units with discounts $\{\gamma_1, \ldots, \gamma_N\}$, both $Z^{-1}$ and $\mathcal{L}^{-1}$ can be approximated with a *linear transformation* $\mathbf{L}^{-1} : \mathbb{R}^N \rightarrow \mathbb{R}^{T+1}$, which takes as input a $\gamma$-space vector with N components and returns as output a discrete $\tau$-space vector until a horizon T:

$$\mathbf{L}^{-1}\big[V_{\gamma_1}(s_t), \ldots, V_{\gamma_N}(s_t)\big] = \big[E[r_{t+0}|s_t], \ldots, E[r_{t+T}|s_t]\big] \tag{9}$$

Approximating $Z^{-1}$ and $\mathcal{L}^{-1}$ with the linear decoder $\mathbf{L}^{-1}$ makes them highly neurally-plausible operations, which we exploit in the rest of the paper. We discuss various $\mathbf{L}^{-1}$ in Appendix A.

## 3.2 TD learning with varying temporal discount and reward sensitivity

Next, we assume that units have different *reward sensitivities* $f_h(r)$, in addition to having different $\gamma$'s. Thus, instead of responding in proportion to $r$, unit $h$ responds according to $f_h(r)$ (we use the sigmoidal functions in Fig. 2a in the rest of the paper). The representation of value is now two-dimensional, each unit being labeled by $\{h, \gamma\}$ (Fig. 2b). The TD backups are given by:

$$V_{h,\gamma}(s_t) \leftarrow V_{h,\gamma}(s_t) + \alpha \delta_{h,\gamma}(t) \tag{10}$$

$$\delta_{h,\gamma}(t) = f_h(r_t) + \gamma V_{h,\gamma}(s_{t+1}) - V_{h,\gamma}(s_t)$$

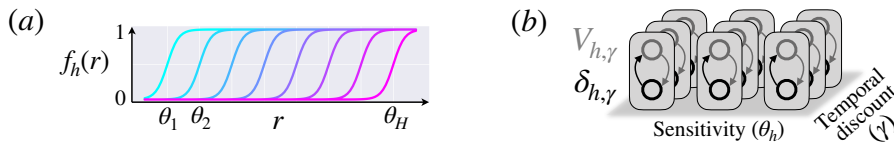

Figure 2: **(a)** Sigmoid reward sensitivities. **(b)** Local TD circuit required by the Laplace code.

In the limit of infinitely steep sigmoidal functions, $f_h(r)$ becomes the Heaviside function $H(r - \theta_h)$, which is 1 if $r > \theta_h$ and 0 otherwise. In this case, the estimate $V_{h,\gamma}(s_t)$ in Eq. 10 converges to:

$$V_{h,\gamma}(s_t) \rightarrow E\left[\sum_{\tau=0}^{\infty} \gamma^{\tau} f_h(r_{t+\tau})\Big| s_t\right] = \sum_{\tau=0}^{\infty} \gamma^{\tau} E\left[f_h(r_{t+\tau})|s_t\right] \tag{11}$$

$$= \sum_{\tau=0}^{\infty} \gamma^{\tau} E\left[H(r_{t+\tau} - \theta_h)|s_t\right] = \sum_{\tau=0}^{\infty} \gamma^{\tau} P\left(r_{t+\tau} > \theta_h|s_t\right)$$

observing that the expectation of the Heaviside function is the definition of the cumulative probability. Comparing this convergence point with Eq. 5, we can proceed as in the previous section and apply, for each $h$, the linear decoder $\mathbf{L}^{-1}$ to the $\gamma$-space vector, obtaining the corresponding $\tau$-space vector. This gives, for each $h$:

$$\mathbf{L}^{-1}[V_{h,\gamma_1}(s_t), \dots, V_{h,\gamma_N}(s_t)] = [P(r_{t+0} > \theta_h|s_t), \dots, P(r_{t+T} > \theta_h|s_t)] \tag{12}$$

Equivalently, applying $\mathbf{L}^{-1}$ to the $\gamma$-space vector with components $V_{h-1,\gamma_i}(s_t) - V_{h,\gamma_i}(s_t)$ returns the $\tau$-space vector with components $P(\theta_{h-1} < r_{t+\tau} < \theta_h|s_t)$. Thus if the resolution along the $h$-dimension is high enough, applying $\mathbf{L}^{-1}$ to the converging points of the TD backups in Eq. 10 recovers $\{P(r_{t+\tau}|s_t)\}_{\tau=0}^{T}$, the set of *probability distributions of immediate rewards at all future timesteps* until T, given that the current state is $s_t$. In the rest of the paper, we will refer to the code provided by $\{V_{h,\gamma}\}$ as a *Laplace code* for value. In Fig. 3 we illustrate the behaviour of the Laplace code in a specific MP, using the smooth sigmoidal functions in Fig. 2a as reward sensitivities.

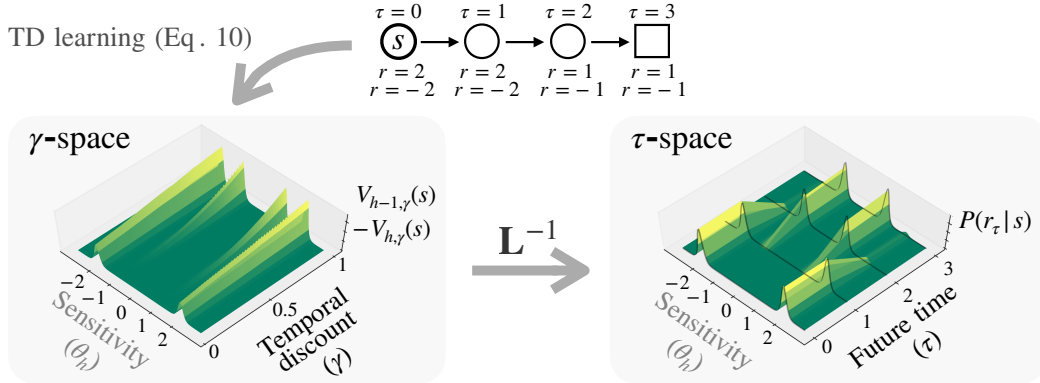

Figure 3: Laplace code applied to a specific MP. **(Top)** Starting from $s$, at timesteps $\tau = 0$ and $\tau = 1$ the agent obtains with $50\%$ probability $r = 2$ and with $50\%$ probability $r = -2$, then at $\tau = 2$ and $\tau = 3$ it obtains $r = 1$ or $r = -1$. **(Left)** Applying the TD backups from Eq. 10 to the MP converges to the set of $V_{h,\gamma}(s)$ shown in the $\gamma$-space, where we plot $V_{h-1,\gamma}(s) - V_{h,\gamma}(s)$ as a function of the $\theta_h$ and $\gamma$ of the unit. **(Right)** The linear decoder $\mathbf{L}^{-1}$ is applied across the $\gamma$-dimension of the $\gamma$-space (see Eq. 12). The same decoding is done for all $\theta_h$'s, filling the $\tau$-space column by column. Using sigmoid $f_h$'s, the final $\tau$-space at time $\tau$ is a smooth approximation to $P(r_\tau = \theta_h|s)$, the probability that the immediate reward in $\tau$ timesteps from the present will be equal to $\theta_h$, given that the current state is $s$. The true (smoothed) immediate reward distributions are shown with thin black lines.

As illustrated in Fig. 3, the Laplace code recovers a temporal map of the problem, indicating all the possible rewards at all future times. We now ask what can we do with this information. If the goal of the agent is to maximize rewards, it needs to be able to compute the *expected Bellman value* of $s_t$ for some particular $\tilde{\gamma}$ defined by the problem. The value of $s_t$ can be computed in the $\tau$-space by:

$$E\left[\sum_{\tau=0}^{\infty} \tilde{\gamma}^{\tau} r_{t+\tau}\Big| s_t\right] = \sum_{\tau=0}^{\infty} \tilde{\gamma}^{\tau} \sum_{r} r P(r_{t+\tau} = r|s_t) \tag{13}$$

The value of $s_t$ can also be computed linearly in the $\gamma$-space, without the need for the $\mathbf{L}^{-1}$ operator:

**Lemma 3.1.** *For infinitely narrowly spaced $\theta_h$'s, the expected Bellman value of $s_t$ satisfies:*

$$E\left[\sum_{\tau=0}^{\infty} \tilde{\gamma}^{\tau} r_{t+\tau}\Big| s_t\right] = \sum_{h=1}^{H} \left(V_{h-1,\tilde{\gamma}}(s_t) - V_{h,\tilde{\gamma}}(s_t)\right)\theta_h \tag{14}$$

*Proof.* See Appendix B. $\qquad\qquad\qquad\qquad\qquad\qquad\qquad\qquad\qquad\qquad\qquad\qquad\qquad\qquad\qquad$ $\square$

In addition to the expected value of states, the $\tau$-space can recover the *value distribution* of states in environments with a single reward per episode.

**Lemma 3.2.** *In MPs with non-zero rewards only in transitions to absorption, if absorption happens in less than $T$ timesteps from all states, the value distribution satisfies:*

$$P\Big(\sum_{\tau=0}^{T}\tilde{\gamma}^{\tau}r_{t+\tau}=V\,\Big|s_t\Big)=(1-\delta_{(V,0)})\sum_{\tau=0}^{T}P\big(r_{t+\tau}=\tilde{\gamma}^{-\tau}V\big|s_t\big) \qquad (15)$$

*where $\delta_{(V,0)}$ is the Kronecker delta.*

*Proof.* See Appendix C. ☐

Examples of this class of environments are Chess, Go, Pong, Mountain Car, Capture the Flag, navigation tasks with one goal per episode, etc. Also, most neuroscience experiments have a single goal per episode. In these environments, our local code recovers not only the value distribution but also the temporal evolution of rewards $\{P(r_{t+\tau}|s_t)\}_\tau$. As we illustrate in section 4, this greatly increases the representational capacity of the code and allows for temporally flexible learning. For example, if the time horizon of the problem shortens from $T$ to $T'$, the agent can immediately recompute the new value distribution using Eq. 15, by simply summing until $T'$ instead of $T$.

### 3.3 TD learning with varying temporal discount, reward sensitivity and memory

The temporal evolution of rewards recovered by the Laplace code (i.e. $\{P(r_{t+\tau}|s_t)\}_\tau$) is essentially a different object from the value distribution (i.e. $P(\sum_{\tau=0}^{\infty}\gamma^{\tau}r_{t+\tau}|s_t)$). In environments with one reward per episode, the temporal evolution *contains* the value distribution, as shown by Lemma 3.2. However, in environments with multiple rewards per episode, the temporal evolution may not contain the value distribution. This is because the temporal evolution recovers the reward distribution at *each* future time, but not their correlations across time. If rewards are temporally correlated, the distribution of the *sum* of rewards across time cannot be recovered from the distribution at each time. A first approach to solve this problem is to deliver the cumulative return obtained in the episode only after termination. Although this indeed allows to recover the value distribution from the temporal evolution in *any* environment using Lemma 3.2, it loses the temporal flexibility. For instance, the agent would no longer be able to immediately adapt to a change in the time horizon.

To approach this problem while preserving temporal flexibility, we extend the ensemble in one more direction, endowing units with different *lengths of memory* $n$. Each estimate $V_{h,\gamma,n}(t)$ is updated at time $t+n$ with a 1-step TD backup using $f_h(R_t^n)$, where $R_t^n = r_t + \ldots + \tilde{\gamma}^n r_{t+n}$ (for now, considering only the single $\tilde{\gamma}$ defined by the problem). The backups at time $t+n$ are:

$$V_{h,\gamma,n}(s_t) \leftarrow V_{h,\gamma,n}(s_t) + \alpha\delta_{h,\gamma,n}(t+n) \qquad (16)$$
$$\delta_{h,\gamma,n}(t+n) = f_h(R_t^n) + \gamma V_{h,\gamma,n}(s_{t+1}) - V_{h,\gamma,n}(s_t)$$

The backups continue for $n+1$ steps after reaching a terminal state, implemented in a computationally efficient way similar to the reverse replay observed in hippocampal place cells [15] (see Appendix D.1 for details). Following the same steps as before, if $f_h(R_t^n)$ is a Heaviside function:

$$\mathbf{L}^{-1}\big[V_{h,\gamma_1,n}(s_t),\ldots,V_{h,\gamma_N,n}(s_t)\big] = \big[P(R_{t+0}^n > \theta_h|s_t),\ldots,P(R_{t+T}^n > \theta_h|s_t)\big] \qquad (17)$$

We illustrate the recovery of $\{P(R_\tau^{n=2}|s)\}_\tau$ for a specific MP in Fig. 4.

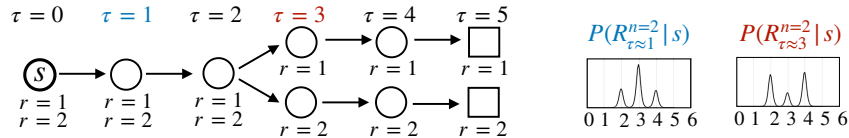

Figure 4: After applying $\mathbf{L}^{-1}$ to the set of $\{V_{h,\gamma,n=2}(s)\}$, reading across the $\tau$ dimension looks closer or further away into the future ($\tau \approx 1$ and $\tau \approx 3$ are shown, the '$\approx$' comes from the regularization factor in $\mathbf{L}^{-1}$, which decreases the precision of the $\tau$-space as $\tau$ increases). Then, reading from $n=2$ recovers the distribution of the discounted sum of $n=2$ consecutive rewards starting at $\tau$.

The $\{h,\gamma,n\}$ code can adapt to changes in the temporal horizon simply by reading from a different $n$, but it can only recover the value distribution for the temporal discount $\tilde{\gamma}$ in $R^n$. In Appendix D.2 we discuss the forward view of a continuous representation of the memory dimension, similar to eligibility traces, that recovers the value distribution for any $\tilde{\gamma}$ without increasing dimensionality.

# 4 Representational power and temporal flexibility of the Laplace code

The Laplace code has a cost: it requires order $N^3$ units compared to order N for the Expectile code (see Appendix E.2 for a more detailed complexity analysis). As we have seen, however, the Laplace code has the advantage of being learnable with a simple and local variation of the traditional TD learning rule. In this section we discuss two additional advantages. The first is *representational power*: by coding for the entire temporal evolution of rewards, the code represents separately states with the same value distribution but different temporal evolution. The second is a form of *model-based temporal flexibility*, allowing for immediate adaptation to changes in the environment.

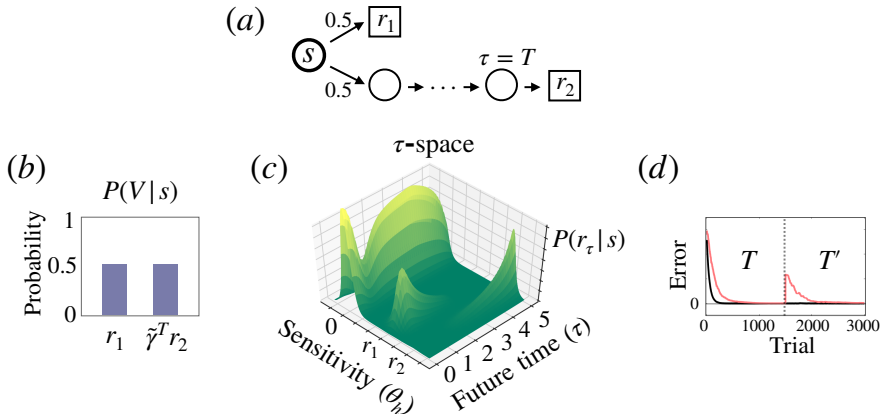

Figure 5: **(a)** MP with parameters $r_1$, $r_2$ and $T$. **(b)** Value distribution at $s$. **(c)** Laplace code estimation of the $\tau$-space for $r_1 = 0.6$, $r_2 = 1$ and $T = 5$. **(d)** Quadratic error between the estimates of the Expectile code (red) or the Laplace code (black) and the true estimates at $s$ (true expectiles and true $\gamma$-space, respectively). In the trial $1500$ the horizon of the task changed from $T$ to $T' < T$.

Consider the MP shown in Fig. 5a. From the starting state $s$, the agent can obtain a temporally close reward $r_1$, or a temporally far reward $r_2$ at $T$. In Fig. 5b we show the value distribution at $s$. Importantly, the value distribution alone cannot disambiguate the parameters $r_1$, $r_2$ and $T$. In other words, it is not possible to recover the value of the parameters from the value distribution, since there exist multiple parameter values that lead to the same value distribution. For example, shortening $T$ to $T-1$ while decreasing $r_2$ to $\tilde{\gamma} r_2$ has no effect on the value distribution. Thus an agent that only learns the value distribution at $s$ cannot recover the values of $r_1$, $r_2$ and $T$ in this MP. On the other hand, the Laplace code learns the temporal evolution of immediate rewards, as shown in Fig. 5c[2]. The temporal evolution separately represents reward magnitude and reward timing, allowing the code to recover the parameters $r_1$, $r_2$ and $T$. Although parameter recovery is evidently not required to infer the value distribution, future work can explore whether the increased representational power is advantageous in the context of function approximation, rather like distributional TD itself.

The temporal evolution of immediate rewards in Fig. 5c can also be used in model-based computations to provide flexibility to changes in the task. For example, imagine that the temporal discount that defines optimal prediction and control in the MP from Fig. 5a changes from $\tilde{\gamma}$ to $\tilde{\gamma}'$, or the temporal horizon shortens from $T$ to $T' < T$, withdrawing the reward $r_2$. The Laplace code (without the memory extension) can immediately recompute the new expected value of $s$ after these changes using Eq. 13 (note that this is true for any MP, and not only for MPs with one reward per episode). Also, it can compute the new value distribution of $s$ after these changes using Eq. 15 (this would require the memory extension in problems with multiple rewards per episode). In Fig. 5d we compare flexibility to a horizon change between the Laplace code (black) and the Expectile code (red), whose estimates need to re-converge to the new value distribution at $s$ under the new horizon $T'$.

# 5 Comparison with the Expectile code

One of strongest piece of evidence in support of the Expectile code in Dabney et al.'s study [10] comes from the decoding of the reward distribution from VTA responses with a decoder designed for

an Expectile code. At first, this result would seem to rule out the Laplace code we have presented here. Indeed, if the VTA uses a Laplace code, a decoder that wrongly assumes an Expectile code would be expected to perform poorly. We show in this section that, in fact, an expectile decoder applied to the Laplace code does recover a good approximation to the encoded distribution, provided the tuning curves of the neurons that code for reward magnitude are adapted to the reward distribution. The idea that the distribution of tuning curves should be adapted to the stimulus distribution is central to several theories of optimal coding [16–18]. We assume here a simple model in which the steep part of the sigmoidal curves in Fig. 2a are positioned around the rewards that are often received.

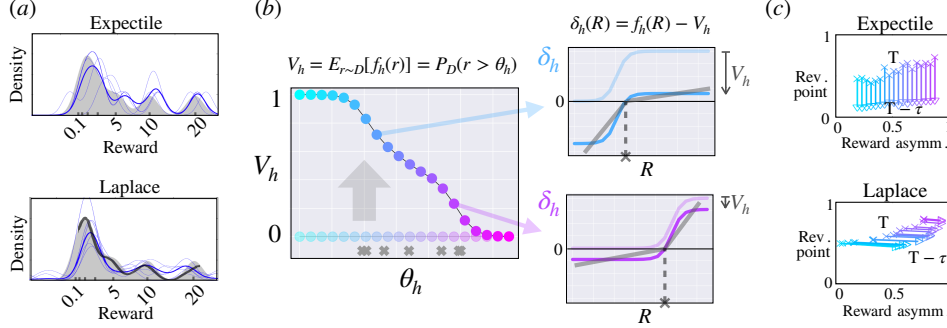

Figure 6: **(a)** The true (smoothed) reward distribution from [19] is shown in shaded grey. In the upper plot value was coded with an Expectile code, in the lower one with an adaptive Laplace code. Thin blue lines show individual decoded distributions assuming an Expectile code, thick blue line shows their mean. The thick black line shows the decoded distribution assuming a Laplace code (the edges are missing due to the adaptive nature of the $\theta_h$). **(b)** Illustration of the correlation between the reversal point and the reward asymmetry for the Laplace code. Left plot shows $V_h$'s before (transparent) and after (opaque) learning, right plots show RPE as a function of reward magnitude for two units, before and after learning. **(c)** Each line corresponds to the RPE of a single unit, at the expected reward time (crosses) and $\tau$ time steps before the expected reward time (triangles). In the Laplace code color is $\theta_h$ and, in the Expectile code, $\tau_i$.

In Dabney et al. [20], the decoder relies on assigning a specific expectile to each unit/neuron. Using the RPE response as a function of reward magnitude, expectile $\tau_i$ corresponds to the *reward asymmetry* of the RPE (i.e. $\alpha_i^+/(\alpha_i^+ + \alpha_i^-)$, see Eq. 3); and the value of that expectile $V_i$ is given by the *reversal point*, i.e. reward magnitude at which the RPE changes its sign. Classifying units like this, the authors were able to decode with reasonable accuracy the reward distribution from [19], where the animals receive random rewards whose amplitude can take one of 7 equally probable values (tick marks in Fig. 6a). To be precise, Dabney et al. did not decode the original distribution but the original distribution smoothed through a Gaussian kernel (shown in shaded gray in Fig. 6a). An example of this decoding from expectile simulations is shown in Fig. 6a (blue lines, upper plot).

However, similar decoding results can be obtained from a Laplace code[3]. We trained an ensemble of Laplace units to encode the same reward distribution and decoded the smoothed reward distribution *assuming that the units actually code for expectiles*. As in Dabney et al., we assumed that the reward asymmetry of the RPE codes for expectile $\tau_i$; and the reversal point of that unit codes for $V_i$, the value of that expectile. The results (blue line, lower plot, Fig. 6a) show that we can decode the reward distribution from [19] with an accuracy comparable to what is obtained when decoding an actual Expectile code for the same distribution (blue line, upper plot, Fig. 6a). Note also that the distribution decoded under the wrong assumption of an Expectile code is only marginally worse than the one recovered by a decoder that rightly assumes a Laplace code (black line, lower plot, Fig. 6a) (see Appendix E.5 for details). We observed similar results for the variable-probability task from [19] (shown in Appendix E.4), that varies reward probability instead of reward magnitude.

This result is due to the correlation between reversal point and reward asymmetry, which is shared by the Laplace and Expectile codes. In the left plot of Fig. 6b, we show the $V_h$'s before and after learning. After learning, $V_h$ converges to $E_{r\sim D}[f_h(r)]$, which is a smooth approximation to $P_D(r > \theta_h)$. In

the upper-right plot of Fig. 6b, we show the RPE of a low-$\theta_h$ unit before and after learning. For a reward magnitude $R$, the RPE is $\delta_h(R) = f_h(R) - V_h$. Since $V_h$ for this unit is high, the tuning curve moves down a large amount, which causes a low reward asymmetry ($\alpha^- > \alpha^+$, compare positive and negative slopes of the linear fit) and a low reversal point (cross on x-axis). On the other hand, the high-$\theta_h$ unit in the lower-right plot ends up having high reward asymmetry ($\alpha^+ > \alpha^-$) and high reversal point. Thus the Laplace code has a similar correlation between reversal point and reward asymmetry as the Expectile code. Furthermore, the reversal points end up being close to the $\theta_h$ of the unit. Since adaptive coding leads to the distribution of $\theta_h$ to resemble the reward distribution, which also resembles the expectile distribution, we see the similar decoding behaviour in Fig. 6a.

Therefore, decoding neural activity in this reward magnitude experiment [19] does not allow us to discriminate between Expectile and Laplace codes. However, these codes can be distinguished with a different type of test. For the Expectile code, reward asymmetry is a neuron-specific feature which specifies the expectile encoded by that neuron, which is the same at all times. By contrast, for the Laplace code, the *reward asymmetry is not an intrinsic property of the neuron* but a consequence of the downward displacement of the response curve (see Fig. 6b), which depends on $V_h$ through $\delta_h(R) = f_h(R) - V_h$. For the Laplace code, any manipulation of $V_h$ only moves the function $\delta_h(R)$ vertically, affecting the reward asymmetry; but for the Expectile code it changes the functional form of $\delta_i(R)$, such that it maintains its reward asymmetry while changing its reversal point $V_i$. Thus, unlike the Expectile code, the Laplace code predicts that the RPE tuning curve $\delta_h(R)$ should have the same functional form $f_h(R)$ for expected and unexpected rewards, but shifted vertically an amount $V_h$ (with possible small changes due to adaptive coding). This prediction was indeed present in the experimental results from [19]. Another test for this prediction is presenting a one-off reward earlier than it is normally expected, say at time $T - \tau$ where $T$ is the usual delivery time. In this case, $V_h(s_{T-\tau}) = \gamma^\tau V_h(s_T)$. For the Expectile code, this has no impact on the reward asymmetry (Fig. 6c, upper). For the Laplace code, the reward asymmetry increases (Fig. 6c, lower). Note that the nearly reverse predictions hold for the reversal point if one uses the $f_h$'s of Fig. 2a.

## 6 Recovering the Laplace code from the successor representation

The Laplace code has non-interacting representations for the reward magnitude $h$ and the temporal discount $\gamma$. As we are about to see, this allows us to think of the $\{V_{h,\gamma}\}$ (considering $n = 0$ for now) as a set of linear projections from an ensemble of systems computing the SR with different temporal discounts. The SR measures the discounted frequency with which all states will be visited given that one starts at a given state [21]. Formally, the $SR^\gamma(s) \in \mathbb{R}^{|S|}$ vector has $s'$ component equal to:

$$[SR^\gamma(s)]_{s'} = E\left[\sum_{\tau=0}^\infty \gamma^\tau \delta_{(s', s_{t+\tau})} \Big| s_t = s\right] = \sum_{\tau=0}^\infty \gamma^\tau P(s_{t+\tau} = s' | s_t = s). \quad (18)$$

Consider the convergence points of $V_{h,\gamma}$ in Eq. 11. Marginalizing over states each term of the sum:

$$V_{h,\gamma}(s_t) \to \sum_{\tau=0}^\infty \gamma^\tau \sum_s P(s_{t+\tau} = s | s_t) P(r_{t+\tau} > \theta_h | s_t, s_{t+\tau} = s) \quad (19)$$

For Markovian states, the last term is time-independent, equal to $P(r > \theta_h | s)$. Therefore:

$$V_{h,\gamma}(s_t) \to \sum_s \left(\sum_{\tau=0}^\infty \gamma^\tau P(s_{t+\tau} = s | s_t)\right) P(r > \theta_h | s) = SR^\gamma(s_t) \cdot \boldsymbol{r}_h \quad (20)$$

where the vector $\boldsymbol{r}_h \in \mathbb{R}^{|S|}$ has components $[P(r > \theta_h | s)]_s$. Thus the set of $\{V_{h,\gamma}(s)\}$ can be computed by a set of linear projections from an ensemble of vectors computing $SR^\gamma(s)$ with different $\gamma$'s, as illustrated in Fig. 7. The same computation can be done for memory $n > 0$, recovering $\{V_{h,\gamma,n}(s)\}$ by projecting $SR^\gamma(s)$ onto the vector $\boldsymbol{r}_{h,n}$ with components $[P(R^n > \theta_h | s)]_s$.

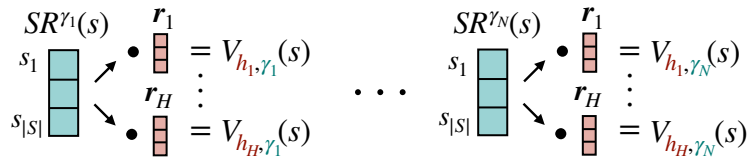

Figure 7: Recovery of the $\gamma$-space $\{V_{h,\gamma}\}$ from an ensemble of SRs with multiple $\gamma$'s.

# 7 Discussion

We developed a distributional RL code for value that has two major advantages over the one recently proposed by Dabney et al. [10]: *(1)* it allows the agent to recover not only the value distributions but also the temporal evolution of the reward distribution; and *(2)* it can be learned with a local learning rule, as opposed to the rule of Dabney et al. which requires that a unit knows the states of other units to update its parameters. The former property increases the representational capacity of the code and allows for temporally flexible decision making, for example by immediately adapting to changes in the temporal horizon. The latter property is appealing and realistic from a biological point of view and leads to a significantly simpler convergence analysis compared to non-local codes, for which convergence needs to be proven in a distributional sense (see for example [20]). Finally, we showed that our distributional code can be computed linearly from an ensemble of systems computing the successor representation with different temporal discounts, a model proposed for the hippocampus [11, 22].

Our code decomposes value distributions and prediction errors across three separated dimensions: reward magnitude, temporal discounting and time horizon. For the first two of these, dopamine neurons in the VTA apparently code for varying reward magnitudes [19]; and it has been proposed that value functions and/or TD prediction errors for different values of $\gamma$ are arranged in a spatially organized manner along the dorso-ventral axis of the striatum [23, 24] (with serotonergic activity 'choosing' which is relevant at a given time [25, 26]). Our work shows that such arrangement is sufficient to recover distributional value information through the inverse Laplace projection. Regarding the third dimension, it is still unclear if time horizon is separated from temporal discount in the dopamine system, but some experimental results suggest a discrete coding of temporal horizon [27].

Our code proposes an indirect approach to recover the statistics of the value distribution, by first recovering the temporal evolution of the immediate reward distribution. Essentially, our code builds a 'temporal map' of possible future rewards, which can be used for model-based computations. This approach could be generalized to other variables beside reward. For instance, if neurons were tuned to spatial location, with tuning curves $f_h(x)$, the inverse Laplace operator would recover a temporal map of future positions. Indeed, in a more general framework the convergence points of the Laplace code are a form of General Value Function [28], which has been proposed as a unifying system to learn about many different variables from the same line of experience [29]. This possibility is consistent with recent results showing that dopamine neurons respond not only to predicted reward but also to other variables such as distance to reward, movement and behavioural choices [30–33].

An important question is whether the reward sensitivities $f_h(r)$ measured in dopamine neurons resemble the sigmoids that we used in our analysis. Dopamine neurons show Hill-like responses to increasing reward magnitude [19], which are indeed sigmoid-like in the logarithmic space [34]. Beyond this similarity, it is important to note that the ideas used to develop the Laplace code are not exclusive to sigmoidal reward sensitivities. For any $f_h(r)$, the Laplace code ends up learning the temporal evolution of the expectation $E[f_h(r)]$ (namely, $\{E[f_h(r_\tau)]\}_{\tau>0}$). As a simple example, if rewards always occur at the same time, and they are distributed according to the distribution $D$ (as in the variable-reward magnitude task from [19]), then $V_h$ in the Laplace code effectively learns $E_{r\sim D}[f_h(r)]$, the expected value of $f_h(r)$ when rewards are distributed as $D$. Thus even if the $f_h(r)$ are not sigmoids, the agent could still use the set of $V_h$'s to decode the distribution $D$, by solving the inverse problem of recovering $D$ from a set of expectations under $D$ (namely, the set of $E_{r\sim D}[f_h(r)]$ for different $f_h$'s).

Rather like the Expectile code, our code can be implemented using function approximation methods like Deep RL networks simply by adding it to the last layer. Combined with function approximators, the additional error signals provided by our code could allow the system to learn richer representations than traditional RL [7], and possibly even richer than those learnt using an Expectile code, since hidden representations now must distinguish between states with the same value distribution but different temporal evolution of the immediate reward distribution. The increased representational power, even without the memory extension, together with our code's locality, will likely give rise to different dynamics of convergence in RL networks, a question we leave open for future work.

## 8 Broader Impact

Understanding human decision-making in uncertain environments is critical not only for neuroscience and psychology but also to address economic, social and medical problems. In various such contexts, humans can benefit from recovering not only the probability distribution of possible events but also the specific times at which they are likely to occur. Here we propose a framework to solve this conundrum, showing how a distributional map of future events can be recovered from a simple extension of traditional reinforcement learning neural models.

## Acknowledgements

We thank Christopher Gagne, Adam Lowet, Nisheet Patel, Reidar Riveland and Naoshige Uchida for fruitful discussions.

## Funding disclosure

A.P. and P.T. were supported by SNF grant #315230_197296. P.D. was supported by the Max Planck Society and the Alexander von Humboldt Foundation.

The authors have no conflict of interests.

## Footnotes

[2]To ease intuition we show the $\tau$-space in Fig. 5c, but the $\gamma$-space has the same representational capacity.

[3]Since in this task RPEs are studied at reward time, dopamine predictions for $V_{h,\gamma,n}$ are independent of $n$ and $\gamma$ (in a notionally absorbing MP), so we note $V_{h,\gamma,n}$ simply as $V_h$.

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
