[Supplementary Material]

# A Local Temporal Difference Code for Distributional Reinforcement Learning

**Pablo Tano**[*]
University of Geneva

**Peter Dayan**
MPI for Biological Cybernetics
University of Tübingen

**Alexandre Pouget**
University of Geneva

# Appendices

## Contents

---

[*]Corresponding author: `Pablo.TanoRetamales@unige.ch`

# A  Approximations of $Z^{-1}$ and $\mathcal{L}^{-1}$

In practice, both the $\mathcal{L}^{-1}$ transform (in continuous-time MPs) and the $Z^{-1}$ transform (in discrete-time MPs) are approximated with a discrete linear decoder $\mathbf{L}^{-1} : \mathbb{R}^N \to \mathbb{R}^{T+1}$ that takes as input a vector representing the $\gamma$-space (with N discrete $\gamma$'s) and returns as output the vector representing the $\tau$-space (with $T+1$ discrete $\tau$'s). In this section, we illustrate $\mathbf{L}^{-1}$ using the Laplace code that varies along $\gamma$ and $h$, so we suggest the reader to visit section 3.2 in the main text before going forward. For the Laplace code that varies along $\gamma$ and $h$, and uses reward sensitivities $f_h$, the $\gamma$-space and $\tau$-space vectors are:

$$\mathbf{L}^{-1}[V_{h,\gamma_1}(s), \ldots, V_{h,\gamma_N}(s)] = \big[E[f_h(r_0)|s], \ldots, E[f_h(r_T)|s]\big] \tag{A.1}$$

where we assume that $s$ is the first state of the MP for notational simplicity (i.e. $s$ occurs at $t = 0$). If $f_h$ is a Heaviside function, as we assume in this section, the recovered $\tau$-space is:

$$\mathbf{L}^{-1}[V_{h,\gamma_1}(s), \ldots, V_{h,\gamma_N}(s)] = \big[P(r_0 > \theta_h|s), \ldots, P(r_T > \theta_h|s)\big] \tag{A.2}$$

Note that the same decoder $\mathbf{L}^{-1}$ is applied independently of the functional form of $f_h$, recovering the $\tau$-space from Eq. A.1 (instead of the $\tau$-space from Eq. A.2). For general sigmoid functions, for example, $E[f_h(r_\tau)|s]$ is the smooth approximation to $P(r_\tau > \theta_h|s)$ illustrated in Fig. 3.

A discrete decoder to approximate $\mathcal{L}^{-1}$ that has been explored in the neuroscience literature [35] is a translation-invariant linear decoder based on Post's approximation to the inverse Laplace transform. However, since this decoder effectively approximates the $n^{\text{th}}$ derivative of the input vector, it is very sensitive to noise. In our framework, the input is often very noisy, since it corresponds to the converging points of different learning traces. In this section we describe two linear decoders that differ from that in [35] and are more noise-resilient. The first linear decoder is a Singular Value Decomposition - based solution to the (regularized) discrete inversion problem, often used for nuclear magnetic resonance data [36]. All the $\tau$-spaces shown in this paper were recovered with this method. The second linear approximation is a translation-invariant linear decoder trained to recover the $\tau$-space from the $\gamma$-space, initialized to the analytical solution from [35].

## A.1  SVD-based approximation

All the $\tau$-spaces shown in this paper were recovered with this method. In a discrete (or discretized) temporal space[2], component $i$ of the $\gamma$-space vector is related to the $\tau$-space vector through:

$$V_{h,\gamma_i}(s) = \sum_{\tau=0}^{T} \gamma_i^\tau P(r_\tau > \theta_h|s) \tag{A.3}$$

where T is the temporal horizon of the $\tau$-space. In matrix form the $\gamma$-space relates to the $\tau$-space through:

$$\boldsymbol{y}_h = F\boldsymbol{p}_h \tag{A.4}$$

where $\boldsymbol{y}_h$ is the vector $[V_{h,\gamma_1}(s), \ldots, V_{h,\gamma_N}(s)]^\top$, $\boldsymbol{p}_h$ is the vector $[P(r_0 > \theta_h|s), \ldots, P(r_T > \theta_h|s)]^\top$ and $F$ is the matrix with components $F_{ij} = \gamma_i^j$. A Tikhonov-regularized solution to the $\tau$-space $\boldsymbol{p}_h$ that corresponds to a certain $\gamma$-space $\boldsymbol{y}_h$ is the $\boldsymbol{p}_h$ that minimizes:

$$||F\boldsymbol{p}_h - \boldsymbol{y}_h||^2 + \alpha^2||\boldsymbol{p}_h||^2 \tag{A.5}$$

where $\alpha$ is the regularization factor. To minimize this expression, we first calculate the SVD of $F$:

$$F = \sum_{i=1}^{L} \sigma_i \boldsymbol{u}_i \boldsymbol{v}_i^\top \tag{A.6}$$

where $\sigma_i$ are the singular values and $\boldsymbol{u}_i$, $\boldsymbol{v}_i$ the left and right eigenvectors, respectively. The ordinary least-square solution to the $\boldsymbol{p}_h$ that minimizes Eq. A.5 is (see [37] for a derivation):

$$\boldsymbol{p}_h = \sum_{i=1}^{L} \Big(\frac{\sigma_i^2}{\alpha^2 + \sigma_i^2}\Big) \frac{\boldsymbol{u}_i^\top \boldsymbol{y}_h \ \boldsymbol{v}_i}{\sigma_i} \equiv \tilde{\mathbf{L}}^{-1} \boldsymbol{y}_h \tag{A.7}$$

Note that the last approximation to the $\tau$-space $(\boldsymbol{p}_h)$ is a linear transformation of the $\gamma$-space $(\boldsymbol{y}_h)$, which we denote $\tilde{\mathbf{L}}^{-1}$. Since $\tilde{\mathbf{L}}^{-1}$ is linear:

$$\tilde{\mathbf{L}}^{-1}(\boldsymbol{y}_{h-1} - \boldsymbol{y}_h) = \boldsymbol{p}_{h-1} - \boldsymbol{p}_h = \left[ P(\theta_{h-1} < r_0 < \theta_h | s), \dots, P(\theta_{h-1} < r_{\mathrm{T}} < \theta_h | s) \right] \quad \text{(A.8)}$$

which is equal to $\left[ P(r_0 = \theta_h | s), \dots, P(r_{\mathrm{T}} = \theta_h | s) \right]$ if the $h$-resolution is high enough. Finally, note that looking at the $\tau$ component of the set of vectors $\{ \boldsymbol{p}_0 - \boldsymbol{p}_1, \dots, \boldsymbol{p}_{H-1} - \boldsymbol{p}_H \}$ defines a probability distribution over the immediate reward at time $\tau$. Thus the last step in our approximation is to normalize the $\tau$-space across $h$ such that, for each $\tau$:

$$\sum_h P(r_\tau = \theta_h | s) = 1 \quad \text{(A.9)}$$

Including the normalization step, the final transformation $\mathbf{L}^{-1}$ connecting $\boldsymbol{y}_{h-1} - \boldsymbol{y}_h$ to $\boldsymbol{p}_{h-1} - \boldsymbol{p}_h$ is:

$$\boldsymbol{p}_{h-1} - \boldsymbol{p}_h = \tilde{\mathbf{L}}^{-1}(\boldsymbol{y}_{h-1} - \boldsymbol{y}_h) \odot \frac{1}{\sum_h \tilde{\mathbf{L}}^{-1}(\boldsymbol{y}_{h-1} - \boldsymbol{y}_h)} \equiv \mathbf{L}^{-1}(\boldsymbol{y}_{h-1} - \boldsymbol{y}_h) \quad \text{(A.10)}$$

where $\odot$ is the element-wise product, and $\mathbf{L}^{-1}$ is the final transformation that matches column $h$ of the $\gamma$-space with column $h$ of the $\tau$-space. Although the final transformation is no longer linear, it is composed by a linear transformation $\tilde{\mathbf{L}}^{-1}$ followed by a normalization step, both highly neurally-plausible operations. Last, note that Eq. A.3 does not impose any explicit constraint on the distance between contiguous $\gamma$-s in the input vector $\boldsymbol{y}_h$. However, in accordance with the theoretical results, we found in practice that the results were generally better if the $\gamma$'s in the input vector are equidistant in the $\frac{1}{\log \gamma}$ space.

**Normalization step**   The normalization step in Eqs. A.9 and A.10 is crucial for long temporal horizons, since regularization causes the overall magnitude of the recovered $\tau$-space to decrease as $\tau$ increases [3]. Normalization amends the decreasing magnitude problem by making the $\tau$-space to sum to 1 for every $\tau$. In addition to its magnitude, regularization also decreases the temporal precision of the recovered $\tau$-space as $\tau$ increases, an issue that is not solved by the normalization step. However, the loss of precision with $\tau$ reflects a scale-invariant temporal approximation, which has strong support in the neuroscience and psychology literature [35,38].

Figure A.1: **(Top)** From $s$, the agent receives a reward $r = 1$ at $\tau = 1$ and $\tau = 8$, and $r = 0$ at every other time. **(Left)** Applying Eq. 4 to this MP converges to the $\gamma$-space. **(Upper-right)** $\tau$-space at $s$ recovered using $\tilde{\mathbf{L}}^{-1}$ without regularization ($\alpha = 0$). **(Lower-right)** $\tau$-space at $s$ recovered using $\tilde{\mathbf{L}}^{-1}$ with regularization ($\alpha = 10^{-4}$).

Importantly, since the normalization step requires the Laplace code that varies along $\gamma$ and $h$, it is not possible for the Laplace code that only varies along $\gamma$ presented in section 3.1. Therefore, for a code

that only varies along $\gamma$, if an $\alpha > 0$ is used in $\tilde{\mathbf{L}}^{-1}$, it is no longer possible to solve the decreasing magnitude problem. We illustrate the effect of $\alpha$ on the code that only varies along $\gamma$ in Fig. A.1. In this simple unnoisy MP, applying $\tilde{\mathbf{L}}^{-1}$ without regularization (i.e. $\alpha = 0$) perfectly recovers the true temporal evolution of expected rewards $\{E[r_\tau|s]\}_{\tau>0}$ (upper-right plot). Including a regularization factor effectively decreases the magnitude and increases the width of the $\tau$-space proportionally to $\tau$ (lower-right plot). Thus if an agent uses a code that only varies along $\gamma$ to recover information about the $\tau$-space, it should be aware of the modifications induced by the regularization factor.

**Temporal resolution**  The SVD-based approximation to $\mathcal{L}^{-1}$ (in continuous-time MPs) and $Z^{-1}$ (in discrete-time MPs) starts with a discretization of time (Eq. A.3). In discrete-time MPs, the most natural approach is simply to define Eq. A.3 over the same timesteps as defined by the MP, which provides a direct approximation to the (regularized) Z-transform until a temporal horizon T. In other words, if we define the matrix $F$ over $\tau = 0, 1, 2, \ldots, \text{T}$, $\mathbf{L}^{-1}$ recovers the $\tau$-space only at the timesteps that are well defined by the problem (as in Fig. 3 and Fig. A.1). However, another approach is to choose a higher temporal resolution than the one defined by the problem, defining matrix $F$ over $\tau = 0, \Delta t, 2\Delta t, \ldots, \text{T}$ for a $\Delta t < 1$ (as in Fig. 5c), which helps with regularization by smoothing the $\tau$-space across $\tau$. For continuous-time MPs, the SVD-based approximation forces us to choose an arbitrary discretization of time $\Delta t$, which determines the temporal granularity of the recovered $\tau$-space and directly interacts with the regularization factor $\alpha$.

## A.2   Trained linear decoder

The key result from [35] is that the inverse Laplace transform can be approximated by a translation-invariant linear decoder $\mathbf{L}^{-1}$ based on Post's approximation to the inverse Laplace transform. In [35] the decoder was not trained but instead consisted of a fixed set of weights that essentially approximate the $n^{\text{th}}$ derivative of the input vector. We found this method to be very susceptible to input noise. Instead, we trained a translation-invariant linear decoder to recover the $\tau$-space from the $\gamma$-space, initializing the weights of the decoder to a similar solution to the one derived in [35]. Although this method recovers a reasonably good approximation of the $\tau$-space, in practice we found that the SVD-based approximation was more reliable and less computationally expensive.

Figure A.2: The weights of the decoder are trained to minimize the quadratic error between the weighted input and the output for all $h$'s and $\tau$'s (except the tails). The vector on the right is obtained by sweeping the linear decoder along the vector on the left. The same weights $[w_1, \ldots, w_L]$ are used for all $h$'s and $\tau$'s ($L = 5$ is shown in the diagram).

The decoding method is schematized in Fig. A.2. The linear decoder is a vector of length $L$ swept along the $\gamma$ dimension trained to recover the $\tau$-space. Since the transformation is translation-invariant, the same weights can be used for all the $L$-sized windows along the $\gamma$ dimension; as well as for all $h$'s. Importantly, we initialized the weights before training to be Gaussian-wavelets, which is the smooth version of the linear decoder used in [35]. This initialization significantly improves training performance compared to random initialization. In practice, we found good decoding results if *(1)*

$L$ is approximately $10\%$ the size of the input vector; *(2)* the input vector is significantly smoothed before training the decoder; *(3)* during training the gradients receive random $\tau$'s and $h$'s; *(4)* the input to the weights is shifted to be centered at zero; and *(5)* the output is normalized across $h$. Although the true relation between the $\gamma$-space and the $\tau$-space is through $\log \gamma$, in the MPs explored in this paper we found that the decoder also works reasonably well in the linear $\gamma$-space. Finally, note that the tails of the $\tau$-space are lost with this decoding method. Since each tail has length $L/2$, the tail behaviour may be problematic if the size of $L$ is not a small fraction of the input vector.

As in the SVD-based approximation, the nature of trained decoder depends on whether time in the MP is discrete or continuous. For discrete-time MPs, a natural approach is to use as a target $\tau$-space a vector defined over the same timesteps as in the discrete MP (i.e. using $\Delta \tau = 1$). An alternative, which applies both to discrete and continuous-time MPs, is to train the decoder using an arbitrary timestep $\Delta \tau$ in the target $\tau$-space vector. In discrete-time MPs, we found that choosing a smaller $\Delta \tau$ than the one defined by the problem often helps with regularization. Here, the target function used to train the decoder was a square-wave function of period $\Delta \tau$, that takes the value of $P(r_{t+\tau} > \theta_h | s_t)$ over the wave length of a square wave function of period $\Delta \tau$. Note that the square-wave function differs from the *actual* target function that should formally correspond to the $\tau$-space, which is a Dirac-comb function equal to $E[f_h(r_\tau)|s]$ in Dirac-like intervals at the timesteps defined by the MP, and equal to zero at all other times. We used the square-wave function because, in practice, it had more noise-resilient behaviour than the Dirac-comb function.

## A.3 Necessity of $\mathbf{L}^{-1}$

The $\mathbf{L}^{-1}$ is not a local operation. However, it is important to note that applying $\mathbf{L}^{-1}$ is not required for our code to converge, so the $\mathbf{L}^{-1}$ operator should not be interpreted as a part of the TD backups of the Laplace code. This contrasts the imputation step from the Expectile code, which is required for convergence. Instead, the $\mathbf{L}^{-1}$ operator should be interpreted as a tool that we can apply after convergence in order to change from the $\gamma$-space to the $\tau$-space, where certain operations are easier to compute. As we show on the main text, however, several quantities of interest in the RL problem can be easily recovered directly from the $\gamma$-space, without the need of applying $\mathbf{L}^{-1}$. For example, Lemma 3.1 shows that the Bellman expected value of a state can be recovered from the $\gamma$-space. Also, note that if $\tilde{\gamma} = 1$, Eq. 11 tells us that $V_{h-1,\tilde{\gamma}} - V_{h,\tilde{\gamma}} = \sum_\tau P(r_{t+\tau} = \theta_h)$. Note that this convergence values equals the right-hand side of Eq. 15 (after adding the Kronecker delta). Thus if $\tilde{\gamma} = 1$, we can recover the value distribution directly from the $\gamma$-space, without the need of changing to the $\tau$-space. In a similar vein, it is likely that most of the information about the $\tau$-space could be reliably recovered from the $\gamma$-space by domain-dependent decoders adapted to the nature of the noise in each domain, a question we leave open for future work.

# B Proof of Lemma 3.1

If all possible rewards are contained in a finite set, an ensemble $\{V_{h,\tilde{\gamma}}\}$ with arbitrarily high resolution in the $h$-dimension satisfies:

$$E\Big[\sum_{\tau=0}^{\infty}\tilde{\gamma}^{\tau}r_{t+\tau}\big|s_t\Big] = \sum_{h=1}^{H}\big(V_{h-1,\tilde{\gamma}}(s_t) - V_{h,\tilde{\gamma}}(s_t)\big)\theta_h \tag{B.1}$$

*Proof.* Let $\{r_1,\ldots,r_M\}$ be the ordered set of all possible rewards that the agent can possibly obtain. Let $\{\theta_1,\ldots,\theta_H\}$ be the ordered set of thresholds of the Heaviside functions, such that for all pairs of rewards in the reward set, there exist at least one $\theta_h$ between them. Formally:

$$\forall i \in [2, M-1]\ \exists \theta_h, \theta_{h-1} \in \{\theta_2,\ldots,\theta_{H-1}\} \ / \ r_{i-1} < \theta_{h-1} < r_i < \theta_h < r_{i+1} \tag{B.2}$$

and $\theta_1 < r_1$, $\theta_H > r_M$. Under these conditions:

$$P(r = r_i) = P(r > \theta_{h-1}) - P(r > \theta_h) \tag{B.3}$$

Furthermore, note that $P(r > \theta_{h-1}) - P(r > \theta_h) = 0$ if there is no $r_i$ between $\theta_{h-1}$ and $\theta_h$. Therefore:

$$\sum_{i=1}^{M} P(r = r_i) = \sum_{h=1}^{H}\big(P(r_\tau > \theta_{h-1}) - P(r_\tau > \theta_h)\big) \tag{B.4}$$

Also, note that for arbitrarily narrowly spaced $\theta_h$'s, if $\theta_{h-1} < r_i < \theta_h$ we can make $r_i$ arbitrarily close to $\theta_h$. Therefore:

$$\sum_{i=1}^{M} r_i\, P(r = r_i) = \sum_{h=1}^{H} \theta_h \big(P(r_\tau > \theta_{h-1}) - P(r_\tau > \theta_h)\big) \tag{B.5}$$

Using this identity, we can write the expected Bellman value of state $s$ (at $t = 0$, for notational simplicity) as:

$$
\begin{aligned}
V^{TD}(s) &= E\Big[\sum_{\tau=0}^{\infty}\tilde{\gamma}^{\tau}r_\tau\Big] \\
&= \sum_{\tau=0}^{\infty}\tilde{\gamma}^{\tau}E[r_\tau] \\
&= \sum_{\tau=0}^{\infty}\tilde{\gamma}^{\tau}\sum_{i=1}^{M} r_i\, P(r_\tau = r_i) \\
&= \sum_{\tau=0}^{\infty}\tilde{\gamma}^{\tau}\sum_{h=1}^{H} \theta_h \big(P(r_\tau > \theta_{h-1}) - P(r_\tau > \theta_h)\big) && \text{(Eq. B.5)} \\
&= \sum_{h=1}^{H} \theta_h \sum_{\tau=0}^{\infty}\tilde{\gamma}^{\tau} \big(P(r_\tau > \theta_{h-1}) - P(r_\tau > \theta_h)\big) \\
&= \sum_{h=1}^{H} \theta_h \sum_{\tau=0}^{\infty}\tilde{\gamma}^{\tau} \big(E[H(r_\tau - \theta_{h-1})] - E[H(r_\tau - \theta_h)]\big) \\
&= \sum_{h=1}^{H} \theta_h \Big(E\Big[\sum_{\tau=0}^{\infty}\tilde{\gamma}^{\tau}H(r_\tau - \theta_{h-1})\Big] - E\Big[\sum_{\tau=0}^{\infty}\tilde{\gamma}^{\tau}H(r_\tau - \theta_h)\Big]\Big) \\
&= \sum_{h=1}^{H} \theta_h\big(V_{h-1,\tilde{\gamma}} - V_{h,\tilde{\gamma}}\big)
\end{aligned}
$$

$\square$

# C Proof of Lemma 3.2

Let all non-absorbing states have rewards equal to zero, let all transitions to absorbing states have non-zero rewards, and let all rewards after absorption be zero. Furthermore, let all paths eventually fall in an absorbing state after a finite number of steps smaller than $T$. Under these conditions, all non-absorbing states satisfy that one (and only one) non-zero reward will be encountered from that state, in less than $T$ timesteps into the future.

For notational simplicity, let all possible rewards be contained in the set $\{1, 2, \ldots, N\}$ (i.e. we are assuming that rewards are discrete and there are at most $N$ distinct rewards). Therefore, from any non-absorbing state $s$, all the possible future events that can happen are the ones illustrated in Fig. C.1, where the squares denote the reward obtained at absorption and circles denote any non-absorbing state. Here $p_{\tau,j}$ is the probability that the agent encounters the non-zero reward $r = j$ exactly $\tau$ timesteps in the future of $t$, which is also equal to the $P(r_{t+\tau} = j | s_t)$ element of the $\tau$-space.

Figure C.1: All possible events from a non-absorbing state in the kind of problems considered in Lemma 3.2. Squares are labeled by the reward obtained in the transition to absorption, and not by the identity of the absorbing state. Circles denote any non-absorbing state.

Consider $\tilde{\gamma} = 1$ for now. Since the agent receives a single non-zero reward per episode (see Fig. C.1), the probability that the sum of rewards across time is equal to a non-zero $V$ is simply the sum of the probabilities of obtaining $V$ at every future timestep, which is obtained by summing the branches in Fig. C.1 that lead to $V$:

$$P\Big(\sum_{\tau=0}^{T} r_{t+\tau} = V \Big| s_t\Big) = p_{1,V} + p_{2,V} + \ldots + p_{T,V} = \sum_{\tau=0}^{T} P\Big(r_{t+\tau} = V \Big| s_t\Big) \tag{C.1}$$

Similarly, if $0 < \tilde{\gamma} < 1$, the probability that the sum of discounted rewards is equal to a non-zero $V$ is obtained by summing the branches that lead to the corresponding discounted reward:

$$P\Big(\sum_{\tau=0}^{T} \tilde{\gamma}^{\tau} r_{t+\tau} = V \Big| s_t\Big) = p_{1,V} + p_{2,\frac{V}{\tilde{\gamma}}} + \ldots + p_{T,\frac{V}{\tilde{\gamma}^T}} = \sum_{\tau=0}^{T} P\Big(r_{t+\tau} = \frac{V}{\tilde{\gamma}^{\tau}} \Big| s_t\Big) \tag{C.2}$$

Last, we need to account for the fact that the sum of rewards cannot be zero, since all absorbing states have non-zero rewards and all paths eventually terminate in an absorbing state. Thus:

$$P\Big(\sum_{\tau=0}^{T} \tilde{\gamma}^{\tau} r_{t+\tau} = V \Big| s_t\Big) = (1 - \delta_{(V,0)}) \sum_{\tau=0}^{T} P\Big(r_{t+\tau} = \frac{V}{\tilde{\gamma}^{\tau}} \Big| s_t\Big) \tag{C.3}$$

$\square$

# D  Details of the memory extension

## D.1  Computational implementation of TD backups for n>0

To implement the backups in Eq. 16 we use a method inspired by the reverse replay observed in hippocampal place cells [15]. Noting the current timestep as $t$ and the termination timestep as $T$, if $t + n < T$, the backup for state $s_t$ is implemented at time $t + n$, before the episode terminates. However, if $t + n > T$ the backup must be implemented after the episode terminates, as explained below.

After termination, the backups continue for $n + 1$ steps. The first backup after termination updates the value of the last visited state, i.e. the state visited when the episode terminated. After this, the second to last state is updated, and so on until reaching the state visited $n$ steps before termination (or, if $n > T$, the first state visited in the episode).

Figure D.1: Temporal line of backups after termination (dotted line).

In Fig. D.1 (top row) we show how the backups take place after termination in an example episode of length $T = 4$, with visited states $s_1 \to s_2 \to s_3 \to s_2$ and obtained rewards $r_{1,\dots,4}$. Consider $n = N \geq 3$. After termination (dotted lines), first we update the value of the last visited state, which was $s_2$. According to Eq. 16, we should update $V_{h,\gamma,N}(s_2)$ using a perceived reward $f_h(R_{t=4}^N)$, which is equal to $f_h(r_4)$ since no reward was obtained after $r_4$. Formally:

$$V_{h,\gamma,N}(s_2) \leftarrow V_{h,\gamma,N}(s_2) + \alpha[f_h(r_4) - V_{h,\gamma,N}(s_2)] \tag{D.1}$$

Then, the value of the second to last state $s_3$ is updated with $f_h(R_{t=3}^N)$, which is equal to $f_h(r_3 + \tilde{\gamma}r_4)$:

$$V_{h,\gamma,N}(s_3) \leftarrow V_{h,\gamma,N}(s_3) + \alpha[f_h(r_3 + \tilde{\gamma}r_4) + \gamma V_{h,\gamma,N}(s_2) - V_{h,\gamma,N}(s_3)] \tag{D.2}$$

followed by

$$V_{h,\gamma,N}(s_2) \leftarrow V_{h,\gamma,N}(s_2) + \alpha[f_h(r_2 + \tilde{\gamma}r_3 + \tilde{\gamma}^2 r_4) + \gamma V_{h,\gamma,N}(s_3) - V_{h,\gamma,N}(s_2)] \tag{D.3}$$

and finally

$$V_{h,\gamma,N}(s_1) \leftarrow V_{h,\gamma,N}(s_1) + \alpha[f_h(r_1 + \tilde{\gamma}r_2 + \tilde{\gamma}^2 r_3 + \tilde{\gamma}^3 r_4) + \gamma V_{h,\gamma,N}(s_2) - V_{h,\gamma,N}(s_1)] \tag{D.4}$$

We now illustrate the backups for a smaller $n$. For instance, if $n = 1$ the backups continue for $n + 1 = 2$ timesteps, shown in black in Fig. D.1 (bottom row). However, it is also possible to continue the process until reaching the first state (shown in grey), analogously to the case $N \geq 3$ but with perceived rewards $f_h(R_t^{n=1})$ instead of $f_h(R_t^N)$.

Note that if $n$ is sufficiently large and $\gamma = 0$ then $\delta_{h,\gamma=0,n}(t + n)$ in Eq. 16 corresponds to a Monte Carlo backup of $V_{h,\gamma=0,n}(t)$ with perceived total return $f_h(R_t^n)$. This agent converges to $E[f_h(R^n)]$ without the need to change to the $\tau$-space. In other words, Monte Carlo distributional backups are included in the Laplace code with the memory extension, using $\gamma = 0$ and $n > T$.

## D.2  Continuous representation of the memory dimension

From the $\{h, \gamma, n\}$ code the agent can adapt to changes in the temporal horizon simply by reading from a different $n$, but it can only recover the value distribution for the temporal discount $\tilde{\gamma}$ in $R^n$.

To recover the value distribution for any temporal discount, a first approach would be to use multiple $R^n$'s with varying temporal discount. However, a continuous representation of the memory dimension is able to recover the value distribution for any $\tilde{\gamma}$ without increasing dimensionality. We present only the forward view of the code, and leave the task of finding a computationally feasible backward implementation for future work. Consider the backups $\Delta V_{h,\gamma,\lambda}(s_t) = \alpha[C_t^{h,\gamma,\lambda} - V_{h,\gamma,\lambda}(s_t)]$, where:

$$C_t^{h,\gamma,\lambda} = \sum_{n=0}^{\infty} \lambda^n f_h(R_t^{n,\gamma}) \tag{D.5}$$

with $R_t^{n,\gamma} = r_t + \ldots + \gamma^n r_{t+n}$ and $\lambda \in (0,1)$. A unit with backups $\Delta V_{h,\gamma,\lambda}(s_t)$ converges to:

$$V_{h,\gamma,\lambda}(s_t) \rightarrow E[C_t^{h,\gamma,\lambda}|s_t] = \sum_{n=0}^{\infty} \lambda^n P(R_t^{n,\gamma} > \theta_h|s_t) \tag{D.6}$$

where we used Heaviside reward sensitivities for the $f_h$'s. Finally we apply the inverse Laplace operator across the $\lambda$-dimension, which leads to:

$$\mathbf{L}^{-1}[V_{h,\gamma,\lambda_1}, \ldots, V_{h,\gamma,\lambda_N}] = [P(R_t^{n=0,\gamma} > \theta_h|s_t), \ldots, P(R_t^{n=T,\gamma} > \theta_h|s_t)] \tag{D.7}$$

Thus this three-dimensional code approximates $P(R_t^{n,\gamma} > \theta_h|s_t)$ for $n = 0, \ldots, T$, the probability that the sum of $n$ contiguous rewards (with temporal discount $\gamma$) is higher than $\theta_h$.

# E Comparison with the Expectile code

## E.1 Comparison with the Local Expectile code

A local variant of the Expectile code is:

$$V_i(s_t) \leftarrow V_i(s_t) + \alpha_i^+ \delta_i(t) \ \text{ if } \delta_i(t) > 0 \tag{E.1}$$

$$V_i(s_t) \leftarrow V_i(s_t) + \alpha_i^- \delta_i(t) \ \text{ if } \delta_i(t) < 0$$

$$\delta_i(t) = r_t + \gamma V_i(s_{t+1}) - V_i(s_t)$$

In this section we will show two important examples in which this code does not learn the value distribution. We will also show the behaviour of our code for the same examples (labeled as Laplace code). In these examples we simulate the local *Quantile* code (instead of the Expectile code), since quantiles are easier to interpret than expectiles. However, all the problems illustrated for the local Quantile code equally apply to the local Expectile code. For a more complete analysis of why the local Quantile/Expectile code fails, we refer the reader to [9].

**Divergent MP** The local Expectile often code fails in MPs in which one state can transition to two or more future states with different reward distribution, a situation that is present in almost all interesting reinforcement learning environments. We illustrate it in Fig. E.1[4]. The true value distribution at $s$ (with temporal discount $\tilde{\gamma} = 1$) has 0.75 probability of $r = 1$ and 0.25 of $r = 2$ (left plot). The local Quantile code (right plot) fails to learn this value distribution. To see this, consider the quantile 0.6, shown in red in the MP. In state $b_1$ this quantile correctly converges to $V_{i=0.6}(b_1) = 2$, since $r = 2$ is the 0.6 quantile of the reward distribution at $b_2$ and no reward is received at $b_1$. In $a_1$ it correctly converges to $V_{i=0.6}(a_1) = 1$. However, when computing the RPEs for state $s$ with the local backup, the quantile 0.6 is updated half of the time with a future $V_{i=0.6} = 1$ (from $a_1$) and half of the time with a future $V_{i=0.6} = 2$ (from $b_1$), so it learns the 0.6 quantile of the reward distribution with equal mass at $r = 1$ and $r = 2$, which is $V_{i=0.6}(s) = 2$. This prediction is incorrect, since the true 0.6 quantile of the value distribution at $s$ is $V_{i=0.6}(s) = 1$.

Figure E.1: MP in which the local quantile code fails to learn the value distribution at $s$. The ground truth shows the true value distribution at $s$ (for $\gamma = 1$). For the quantiles (right panel) the x-axis corresponds to the value of $V_i$ and the y-axis to the fraction of the total number of estimates that converged to that value.

On the other hand, the Laplace code correctly learns the value distribution at $s$ (middle plot). Here, since we are using $\tilde{\gamma} = 1$, we do not need to apply the $\mathbf{L}^{-1}$ operator to recover the value distribution (see Appendix A.3). Also, note that since in this MP there is a single reward per episode, the Laplace code does not need the memory extension.

**Navigation task** The problem illustrated in the previous section compromises the performance of the local Expectile code in a very common RL environment, the navigation task shown in Fig. E.2a. In this task, the agent moves randomly between the squares with the actions up, down, left and right. Grey squares give no reward, and the episode terminates at the coloured squares obtaining a reward sampled from the respective histograms. The agent is then randomly re-positioned in the maze and a new episode starts.

The true value distributions of all the states in this problem are perfectly captured by the Laplace code without the memory extension, as shown in Fig. E.2c (the ground truth smoothed distributions are

not visible since they overlap with the Laplace code for all states). In contrast, in Fig. E.2b we show that the local quantile code fails to learn the value distributions, by overestimating the probability of getting rewards from the green squares.

Figure E.2: **(a)** Navigation problem in which the local quantile code fails to learn the value distribution of states (positions). **(b)** Convergence estimates of the Local quantile code. The x-axis in the histograms corresponds to the value of $V_i$ and the y-axis to the number of estimates that converged to that value. **(c)** Convergence estimates of the Laplace code and true (smoothed) value distributions. Since we use $\tilde{\gamma} = 1$, we do not need to apply the $\mathbf{L}^{-1}$ operator (see Appendix A.3). The curves inside each square correspond to $V_{h-1,\gamma=1}(s) - V_{h,\gamma=1}(s)$ for each position $s$, convoluted with a well-like function $g(x)$ equal to zero if $x \in [-0.5, 0.5]$ and equal to 1 otherwise. This convolution cancels the responses at $r = 0$ and is meant to approximate the Kronecker delta $\delta_{(V,0)}$ in the right-hand side of Eq. 15.

## E.2 Computational complexity of the Laplace code and the Expectile code

Although a detailed comparison of computational complexity between our code and the Quantile/Expectile code is outside the scope of this paper, we highlight some straightforward aspects in this respect. The Laplace code, without the memory extension, requires the computation of $\mathcal{O}(N^2)$ RPEs per timestep and $\mathcal{O}(N^2)$ memory, with $N$ the number of neurons across the $h$ or $\gamma$ dimension (approximated as equal for simplicity). With a memory extension of size $n$, our code requires the computation of $\mathcal{O}(nN^2)$ RPEs per timestep and the update of $\mathcal{O}(n)$ cumulative rewards $R_t^n$; and $\mathcal{O}(nN^2 + |S|n + n)$ memory. In contrast, the expectile code requires to compute $\mathcal{O}(N)$ RPEs per timestep and $\mathcal{O}(N)$ memory, with $N$ the number of expectiles. However, the expectile code requires the imputation step to occur at every timestep, which is a high-dimensional, convex optimization problem [9]. In contrast, the $\mathbf{L}^{-1}$ operator is *not* required for the Laplace TD code to converge, so its complexity should not be taken into account for computing the TD backups.

## E.3 Decoding details

As illustrated in Fig. E.3a, the reversal point was quantified by first finding the higher reward magnitude with negative RPE and the smaller reward magnitude with positive RPE. The reversal point is the intersection between a line connecting these points and the axis $\delta = 0$. The slopes $\alpha_+$ and $\alpha_-$ were computed by applying linear regression independently to the points with positive and negative RPEs (Fig. E.3b).

Figure E.3: **(a)** Computation of the reversal point. Blue circles correspond to different RPEs of one dopamine unit (y-axis) to a certain reward magnitude (x-axis). **(c)** Computation of the reward asymmetry $\frac{\alpha_+}{\alpha_- + \alpha_+}$.

We used the utility function from [39] to quantify the actual reward perceived by the agent. We excluded from the analysis units whose RPE was strictly $\geq 0$ or $\leq 0$ for all reward magnitudes (i.e. units with no reversal point). To quantify reversal point and reward asymmetry we simulated RPEs assuming that the reward perceived by the agent comes from a normal distribution centered at the reward magnitude that was actually received.

The adaptive nature of the $\theta_h$ was modeled simply by sampling the position of the $\theta_h$ from Gaussian distributions centered at the true rewards (weighted by their frequency), with the same width as the grey distribution in Fig. 6a. In the variable-probability task, we assume that the set of $\theta_h$ used to code for rewards depends on the observed cue. We assume that the adaptation process converges to a fairly stable set of $\theta_h$, and that RPEs are measured after convergence.

The inaccuracies in the decoded distributions in Fig. 6a come from two sources. Given a finite set of expectiles there are multiple distributions consistent with them, represented by the thin blue lines (their average is the bold line). In addition to this, in the adaptive code the sparse $\theta_h$'s are normally distributed around the empirical rewards, which represents a second source of noise.

### E.4  Decoding the variable-probability task

In the main text we analyzed decoding for the variable-magnitude task. We decoded the adaptive Laplace code assuming that their reversal points and reward asymmetries corresponded to expectiles. Here we apply the same decoding to the variable probability task from [19]. The details of the decoding procedure are the same as for the variable reward magnitude task.

In this task there are three odors associated with a following water reward of $4.75\mu l$ with different probabilities ($P_1 = 0.1$, $P_2 = 0.5$ and $P_3 = 0.9$). We modeled the task as two discrete states: a cue state followed by a reward state, and studied the responses of $\{V_{h,\gamma,n=0}\}$, with $\gamma \in [0.5, 1]$ (this $\gamma$-range roughly corresponds to the one measured experimentally in [23]). The adaptive $\{\theta_h\}$ used to code for rewards depends on which odor cue is received, they are sampled from $P_i N(\mu_0, \sigma) + (1 - P_i)N(\mu_1, \sigma)$.

Figure E.4: **(a)** Expectile decoding when value is coded with an Expectile code. **(b)** Expectile decoding when value is coded with an adaptive Laplace code with a single $\gamma = 1$. **(c)** Expectile decoding when value is coded with an adaptive Laplace code with multiple $\gamma$'s between 0.5 and 1.

The decoded distribution assuming that the reversal points are reward asymmetries correspond to an expectile code are shown in Fig. E.4, when the true value function is coded with expectiles (Fig. E.4a), when it is coded with an adaptive Laplace code with $\gamma = 1$ (Fig. E.4b), and when it is coded with an adaptive Laplace code with multiple $\gamma$'s between 0.5 and 1 (Fig. E.4c). Since the reward prediction errors are studied at cue time, units with different $\gamma$ make different predictions. However, even with multiple $\gamma$'s we are able to decode with fairly high reliability the correct value distributions assuming an expectile code. We omitted from the model the 'reward equal to zero' that is received at the cue state. Finally, we only used units with $n = 0$, but including units with $n > 0$ does not affect the quality of the decoding.

### E.5  Decoding Laplace code from Laplace code

The black curve in Fig. 6a was decoded by coding the reward distribution with the Laplace code, as in Fig. 6b, and decoding it correctly assuming a Laplace code. To decode the Laplace code, one needs to *(1)* measure the RPE tuning curve to varying reward magnitudes before learning (i.e. $\delta_h(R)$), which is equal to $f_h(R)$ if one assumes that $V_h = 0$ before learning; *(2)* measure the downwards shift of $\delta_h(R)$ after learning; *(3)* the downwards shift is equal to $E_{r\sim D}[f_h(r)]$; and *(4)* solve the inverse problem of recovering $D$ from a set of expectation under $D$ (namely, the set $\{E_{r\sim D}[f_h(r)]\}_h$). For

the sigmoidal $f_h$'s used in the simulations, this expectations are simply cumulative probabilities under the distribution $D$, so step *(4)* is trivial:

$$E_{r \sim D}[f_h(r)] = P_D(r > \theta_h) \tag{E.2}$$

A critical prediction of the Laplace code (implicit in step *(2)*) is that the functional form of $\delta_h(R)$ should remain the same before and after learning, only shifted vertically[5]. This is illustrated in Fig. 6b in the main text. This is not the case for the Expectile code, which predicts a very different functional form of $\delta_h(R)$ before and after learning (in utility space, it predicts a horizontal shift in the tuning curve, which maintains reward asymmetry whille changing the reversal point). Indeed, the experimental results from [19] suggest a vertically shifted $\delta_h(R)$ for expected versus unexpected rewards, supporting the predictions of the Laplace code.

## Footnotes

[2]See the *Temporal resolution* paragraph below for more details on the discretization of time.

[3]This happens because eigenvectors with large $\sigma_i$ tend to have higher activity at shorter $\tau$'s, and eigenvectors with small $\sigma_i$ tend to occupy the larger $\tau$'s. Thus in the final $\boldsymbol{p}_h$, the regularization factor $\frac{\sigma_i^2}{\alpha^2 + \sigma_i^2}$ decreases more the activity at longer $\tau$'s than at shorter $\tau$'s.

[4]We thank Adam Lowet for this example.

[5]Note that the adaptive coding process that affects the $\theta_h$'s could make the $f_h$'s slightly different before and after learning, which should be taken into account.