[Reviews · NeurIPS 2020]

Review 1

Summary and Contributions: A local temporal difference code is introduced which is indexed by reward magnitude, temporal discount, and time horizon. Through this code, an agent can un use only local TD backups to recover the value distribution. The approach is more biologically plausible than other methods for distributional RL (e.g. quantile regression for distributional RL).

Strengths: Strengths of the contributions: No imputation step is required. An agent can separately represent states with the same value distribution but different evolution of immediate rewards. Multiple temporal discounts and multiple time horizons allows the agent to adapt to changes in the environment. An interesting connection to the successor representation is discussed. To my knowledge, the approach presented in this paper is novel. I think it is a solid technical contribution to distributional RL and opens up interesting directions of research.

Weaknesses: Memory requirements are increased (N^3 units are required instead of N). Empirical evaluation seems limited to simple domains and didactic examples. However, I think this OK. I think the writing can be improved (see my comments in the Clarity section of the review).

Correctness: Appendix A: "Note that P(r=r_i) = P(r > \theta_{h-1}) - P(r > \theta_h)." I'm not sure if this statement is true. For instance, suppose there are 3 reward values, r_i, r_j, r_k, which lie between \theta_{h-1} and \theta_h. Then P(r=r_i) = some fraction of (P(r > \theta_{h-1}) - P(r > \theta_h)). However, when N is arbitrarily large, this issue goes away, since the difference between \theta_{h-1} and \theta_{h} can be small enough so that there is only one reward value r_i that lies in between them.

Clarity: The writing was a bit dense. I think it would be helpful to have definitions and theorems stated explicitly / separately. For example, it was not immediately clear to me what "non-locality" means exactly. Why does receiving samples from the imputed distribution mean that it is non-local, given that the imputed distribution is the distribution imputed at the next state (which is local to the current state)? I think that having a preliminaries section where notation and terms are clearly defined (e.g. the definition non-locality) would help the reader. Perhaps it would also be helpful to have the definitions of quantile, expectile, Heaviside function, and Laplace transform stated so that the reader can easily refer back to them. I found Figure 1 a bit hard to parse on a first read. In Figure 1(c), it would be helpful to label the axes. In the caption for Figure 1, the description for (b) is for image (c) and vice versa. Line 33: V is usually the variable used for value function in RL. I think it would be helpful to use a different variable to avoid confusion. Line 136-137: Why would multiple R^n's be required, wouldn't the only thing that would need to be varied be the temporal discount? Having not previously read reference [10], I was a bit lost in Section 4. It also felt like an abrupt transition from Section 3. Line 185: What is VTA?

Relation to Prior Work: It is clearly discussed how this work differs from some previous contributions, but I think this can be expanded. I think a separate related works section should be added. The approach in this work seems like a direct application of generalized value functions (GVFs) (see for example https://openreview.net/pdf?id=rygvZ2RcYm). I think this literature should be cited and the relationship to it discussed.

Reproducibility: Yes

Additional Feedback: Minor (insignificant) point: In Equation 4, on a first read, it appeared that \theta_h was now a function with input V_{h-1, \tilde{\gamma}} - V_{h, \gamma}. For clarity, maybe put the \theta_h after the (V_{h-1, \tilde{\gamma}} - V_{h, \gamma}). On line 108 it was not clear what the notation \delta_{V, 0} means. Update (post rebuttal): I thank the authors for addressing my comments. I keep my score of 6. I think this paper produces a valuable contribution. My main feedback about the paper is that the clarity can be improved. I had a difficult time understanding the significance of the results in Section 4 potentially due to my lack of neuroscience background, and I was able to appreciate the technical contribution in Section 2 because of my familiarity with distributional RL. I imagine that someone not familiar with either distributional RL or neuroscience might be lost when reading this paper. Some ideas for improving clarity: - Include a background section which goes over the neuroscience and distributional RL background required to fully understand the paper. For example, explain the recent work which proposed a model of dopaminergic activity in the brain based upon distributional RL in more detail, and much earlier in the paper. - Include a preliminaries section in which notation and terms are clearly defined, as I described in my review.


Review 2

Summary and Contributions: This paper builds on recent work which proposed a model of dopaminergic activity in the brain based upon distributional reinforcement learning. In this work, the authors propose an alternative learning rule which is ‘local’, in that the learning update depends only on the reward and a single value estimate at different points in time like in standard temporal difference learning. The method could be seen as demonstrating that a form of successor representation can represent the distribution over future returns, in addition to the value function. Its behavior is shown on a handful of small simulation problems. Finally, the authors show that the distribution decoding results of Dabney et al. (2020) are also consistent with their proposed model.

Strengths: This is a very dense paper, but also extremely interesting. It builds on recent work of Momennejad & Howard (2018) showing the capabilities of a multi-scale successor representation to predict future values through the use of the inverse Laplace transform. The authors expand this approach significantly allowing similar techniques to represent the distribution of returns (value distribution), which is a strong contribution in and of itself. On top of this, the proposed model seems able to account for many of the neuroscientific results found in the recent Dabney et al. paper, but with the key advantage of not requiring communication between separate estimators during the learning update. The appendix does a great job of explaining many additional details that are really essential, but would not have fit into a conference length paper.

Weaknesses: As mentioned above, this is a dense paper and perhaps not the most accessible without significant background reading. I don’t think this necessarily needs to be the case, and I’d suggest giving section 2 more space to allow the build up of the model to be more gradual. While not a weakness in the paper, there are some trade-offs in this model compared with the previous work that should be considered. The simulations vary gamma along 300 values, and in the full approach cells must also allow specialization among a range of rewards and temporal horizons (from an RL view I would call these n-step returns, but they are also referred to as amounts of ‘memory’). Some rough back of the napkin math shows why this could be a bit problematic, or at least very suggestive of the type of approximations the brain would need to make if following this model. Rewards are essentially being binned into H different values. Similarly n-step returns for varying n. With just H=20, n=100 and the number of discounts used in the work, we are looking at ~600k different dopaminergic neurons required by this model. To me, this suggests that along one or more of these axis we should have quite conservative values (such as n being relatively small). But this would also limit what can be learned. I don’t view this as a *real* weakness, as it is suggestive of experiments that could be used to distinguish between this work’s method and the one proposed in previous work.

Correctness: I see no issues with correctness in this work, and I went through the proof and other work in the appendix.

Clarity: Despite my comments around density, I thought this was a very well written paper. Additionally, I found the discussion of approximating the inverse Laplace transform in the appendix to be among the better explanations among those found in related work.

Relation to Prior Work: Related prior work is discussed clearly throughout the paper, and the differences of this work are made amply clear. Perhaps the connection with other successor representation work could be improved slightly, as to my mind this builds so heavily on that area.

Reproducibility: Yes

Additional Feedback: The connection between {V_{h,\gamma}}_{h,\gamma} and {Pr(r_{t+\tau})}_\tau could use more explanation. There paper jumps between these two quite often, and readers could benefit from more space devoted to understanding the connections between the value estimates and the probabilistic objects they are related to. I mentioned the, apparent, importance of having many different reward sensitivities (h) and n-step returns (n, temporal horizons) previously. Could the authors discuss this point in their rebuttal? Doesn’t having a fairly small number of n or h significantly limit what can be represented/learned? Where do the values of \theta come from? It would seem that these would need to be consistent (i.e. the same) across all the dopamine neurons in the population in order for these results to hold. It seems like these would need a type of ‘global’ update in order to ensure consistency (perhaps not in the dopamine system, but somewhere?). Could you clarify this a bit? The authors refer to the different reward sensitivities as producing varying levels of optimism, but this seems like it would only be optimistic/pessimistic in terms of immediate rewards. When reward sensitivities are combined with the n-step specialization, then I could see it encompassing the full return as well, but only if n is taken to infinity. ---- Update ----- After rebuttal and discussion with other reviewers I am slightly lowering my score. I feel that this is a significant contribution that should be accepted, but agree with other reviewers that it could be presented better to improve impact on the NeurIPS readership. Despite this, I do think the paper should be accepted, but would strongly encourage the author(s) to look at improving the clarity of the work (agreeing with another reviewer here that these concepts can be presented more clearly).


Review 3

Summary and Contributions: This paper proposes a temporal difference code for distributional reinforcement learning consisting of an ensemble of independent units, which can recover the value distribution via local TD backups. I have read the other reviews and the author's rebuttal. Although seemingly a good technical contribution,I feel this paper is written in a way that is too inaccessible to readers outside of this niche sub-field. Nevertheless, I have increased my score.

Strengths: This paper presents an interesting approach to fully recovering value function distributions via local TD backups of an ensemble of non-interacting units.

Weaknesses: My main concern with this paper is with the clarity of the writing: there are too many references to supplemental material, too many references to external references, and in general the figures and discussion are difficult to follow. This limits the amount of knowledge transfer readers of this paper will receive. I encourage the authors to revise this draft, as it seems to be able to provide a valuable contribution, but in its current form it is failing to do so.

Correctness: Difficult to ascertain with confidence, as the paper is rather difficult to follow as presented. There was also no code provided with the submission, which makes it difficult to verify and reproduce.

Clarity: This is my biggest issue with this paper: it is _very_ difficult to follow and most of the figures are difficult to interpret. In more detail: - Overall, there are too many references to the supplemental material (e.g. "see SM-C") for things that are necessary for understanding the main paper. - Line 76: what is a Heaviside function? - Figure 1 (c): What is the scale of those plots? What do the bar plots on top of the grid represent? - Figure 1 (b): It's not at all clear how to interpret these plots. What are the dark and grey lines on the right plot meant to represent? The shapes don't quite match with the left plot, is this good? bad? unimportant? In summary: what is the reader meant to take away from this plot? - In the figure 1 caption, it says that figure (b) is "for an agent moving randomly in an absorbing grid world", but the main text says it's for "a linear 4-state MDP" (line 94). which is it? - In the caption for Figure 1 (c), it's not at all clear why [V_{h-1,\gamma}(s) - V_{h,\gamma}(s)] is the value to plot. Why is this difference interesting? - It is not clear where equation (3) came from. - It is not clear where equation (4) came from, perhaps state it as a Lemma or Theorem with proof (proof could be in supplemental material)? - Line 104 "where we assume discrete rewards", there is an implicit assumption of finiteness as well. - Line 105: "The set {P(r_{t+\tau})}_{\tau>0}", is this a set of probabilities or a set of distributions? It's not clear. - Line 108: "P(...) = (1 - \delta_{V,0})", is \delta the RPE? The subscripts are different than what was defined previously. - Line 112-113: "which is quantified by V_{h*-1,\gamma=1| - V_{h*,\gamma=1}", it's not clear why this follows. - It is not clear where equation (6) came from. - In equation (7) you've changed the indices of the summation from \tau to `n`. Why not stick with `tau`? - It is not clear where equation (8) came from. - Line 154: "allows the agent to adapt immediately to changes in the environment". How is the agent adapting if there are no decisions to be made? How do you evaluate this? - Figure 3 caption: "quantile code (red) or Laplace code (black)" it is not clear what "quantile code" or "Laplace code" correspond to in the definitions given previously. - Figure 3 (b) and (c) are used to argue that only using \gamma is not enough for recovering the distribution, relative to using the 2-dimensional code, but it's not very clear how to compare and interpret these plots. - Line 158: What are these \delta functions? These seem to have a different type than all the other \deltas used previously. - Line 185: What is VTA? - Line 188-189: "Indeed, if the VTA uses a Laplace code, a decoder that wrongly assumes an expectile code would be expected to perform poorly." Why? - It's not at all clear how to interpret Figure 4 (b) and (c). - Figure 4 (a) caption: "the edges are missing due to the adaptive nature of the \theta_h". It's not clear why this is the case. - Figure 4 (c) caption: What are \alpha^+_h and \alpha^-_h? - Figure 4 (d) caption: "the x-axis ... reward asymmetry", this hasn't been defined, what is it? - Line 215: "reversal point", what is this? it has not been defined. - Figure 5: What are "slow/fast SR(s_t)"? - In the proof in the appendix, it's not clear how the third and fourth equalities follow. - It would be good to make it explicit from the beginning that this paper is dealine only with the fixed-policy case, and not with control, which is traditionally what reinforcement learning is about.

Relation to Prior Work: Prior work is discussed well, but it seems like it is assumed the reader has a lot of prior knowledge from related literature.

Reproducibility: No

Additional Feedback:

[Author Response · NeurIPS 2020]

We are delighted to see that all reviewers recognized the important scientific contribution of our paper. We apologize for the dense writing and agree that we need to heavily edit the final version. As suggested by R2, we will add in particular a **background section**, which will include short description of traditional TD learning, the Laplace transform and the expectile code. The subsections on traditional TD learning and the Laplace transform should clarify Eq 2 and 3, respectively (as requested by R4). The nonlocality of the update rule of the expectile code (which was unclear for R2) will be dealt with explicitly in this section. Briefly, the update rule is non-local because to compute the RPE of unit $i$, unit $i$ needs to know the state of a random unit $j$. This should make it easier to understand Section 4 even if the reader hasn't read [10]. We will also expand Section 2 to make it more gradual, stating theorems and definitions explicitly.

**Figure 1** As noted by the reviewers, we reversed the figure captions for subplots b and c, for which we apologize. To clarify Fig. 1, we will insert the following diagram, which shows how we go back and forth, via the Laplace transform and its inverse, between the $V_{h,\gamma}$'s (the $\gamma$-space; left plot) and the temporal evolution of the immediate reward distribution ($\tau$-space; right plot). The Markov Process in Fig. 1 (b) will then serve as an example for this diagram, clarifying that in the $\gamma$-space we now measure $V_{h-1,\gamma} - V_{h,\gamma}$ (instead of $V_{h,\gamma}$) to recover the distribution $P(r_\tau = \theta_h)$ (instead of the cumulative distribution, $P(r_\tau > \theta_h)$), a question from R4. We hope that this clarifies the connection between $\{V_{h,\gamma}\}_{\gamma \in (0,1)}$ and $\{P(r_{t+\tau} > \theta_h)\}_{\tau>0}$, a concern of R3.

**Range of $h, \gamma, n$ and quality of approximation** R3 expressed the concern that our approach might require an inordinate number of neurons in the Ventral Tegmental Area, where reward prediction errors are believed to be encoded in the brain. To investigate this issue, we explored how the number of units along the $h$-,$\gamma$-,and $n$-dimension govern the quality of the learnt value distribution. In *(a)* we show the normalized error in the $\gamma$-space (i.e. the percentage difference between the estimated and true $V_{h,\gamma}(s)$ distribution) using $|h|$ values of $\theta_h$'s uniformly distributed between $r_1$ and $r_7$ in the variable reward-magnitude task. In general, to represent $P$ rewards with negligible error, we need $|h| \sim 3P$. In *(b)* we show the normalized error as a function of the number of $\gamma$'s (using $|h| = 20$), for the Markov Process shown on the top. About 10 values of $\gamma$ are sufficient, considerably less than the 300 values that we used in the original simulations (this result also holds in the $\tau$-space, recovered with the linear decoder). Finally, in *(c)* we study the effect of $|n|$ in recovering the $V_{h,\gamma,n=99}(s)$ distribution (with $|h| = |\gamma| = 20$), for two different $\tilde{\gamma}$'s, the temporal discount defined by the problem. If $\tilde{\gamma}$ is not too high ($< 0.95$), the rewards received very far away in the future carry almost no weight, so including them (via a larger $|n|$) does not improve the estimated value distribution.

This analysis suggests that about 20 values covering the range along each dimension, for a total of 8000 units, can represent a wide range of problems. However, $|n|$ constitutes a very hard constraint on the range of representable problems, specially if $\tilde{\gamma}$ is high. This constraint softens with the continuous representation of the $n$-dimension (Eq. 8). Some other forms of function approximation could directly reduce the $N^3$ dimensionality, although possibly compromising locality. We will further explore the limits of representable problems in the final version.

**Related work** We agree with R2: the $V_{h,\gamma}$ to which the Laplace code converges can be formalized as a Generalized Value Function with cumulant function $C_t = f_h(r_t)$ and constant termination function $\gamma$. This interpretation opens interesting routes to function approximation, which we will discuss in the final version.

**Code release** We intend to release the code on GitHub as soon as the submissions are no longer anonymous.

**Others** We thank the reviewers for all the detailed comments (in particular R4). We can't respond to all of them here due to space constraints but we will make sure to address them all in the final version.

[Meta-Review · NeurIPS 2020]

The reviewers appreciated the interesting and novel contribution made here. However, Reviewers 2 and 4 expressed some serious concerns about the legibility of the paper. To quote the discussion, "Someone not familiar with either distributional RL or neuroscience will be lost when reading this paper." The question is therefore whether the issue can be resolved during this conference cycle. I believe it can but that it will require significant editing; I also think it is critical to support interdisciplinary work. However, the burden of clarity remains on the authors. Beyond what the reviewers have said, some recommendations: - There is no doubt that the same argument can be made, while moving some parts to the appendix, especially additional discussion points - Introduce the readers to both neuroscience and distributional RL in a necessarily longer backgroudn section (the RL part is woefully short as it stands) Finally, it may be that this conference paper is an advertisement for a larger journal paper, judging by its density.